# Dual-Function Smart Windows Using Polymer Stabilized Cholesteric Liquid Crystal Driven with Interdigitated Electrodes

**DOI:** 10.3390/polym15071734

**Published:** 2023-03-31

**Authors:** Xiaoyu Jin, Yuning Hao, Zhuo Su, Ming Li, Guofu Zhou, Xiaowen Hu

**Affiliations:** 1College of Physics and Electronic Information, Yunnan Normal University, Kunming 650500, China; 2SCNU-TUE Joint Research Lab of Device Integrated Responsive Materials (DIRM), National Center for International Research on Green Optoelectronics, South China Normal University, No. 378, West Waihuan Road, Guangzhou Higher Education Mega Center, Guangzhou 510006, China; 3Solar Energy Research Institute, Yunnan Normal University, Kunming 650500, China

**Keywords:** cholesteric liquid crystal, interdigitated electrodes, transmittance, reflection bandwidth, smart window

## Abstract

In this study, we present an electrically switchable window that can dynamically transmit both visible light and infrared (IR) light. The window is based on polymer stabilized cholesteric liquid crystals (PSCLCs), which are placed between a top plate electrode substrate and a bottom interdigitated electrode substrate. By applying a vertical alternating current electric field between the top and bottom substrates, the transmittance of the entire visible light can be adjusted. The cholesteric liquid crystals (CLC) texture will switch to a scattering focal conic state. The corresponding transmittance decreases from 90% to less than 15% in the whole visible region. The reflection bandwidth in the IR region can be tuned by applying an in-plane interdigital direct current (DC) electric field. The non-uniform distribution of the in-plane electric field will lead to helix pitch distortion of the CLC, resulting in a broadband reflection. The IR reflection bandwidth can be dynamically adjusted from 158 to 478 nm. The electric field strength can be varied to regulate both the transmittance in the visible range and the IR reflection bandwidth. After removing the electric field, both features can be restored to their initial states. This appealing feature of the window enables on-demand indoor light and heat management, making it a promising addition to the current smart windows available. This technology has considerable potential for practical applications in green buildings and automobiles.

## 1. Introduction

Smart windows have gained popularity in recent decades due to their ability to control indoor solar irradiation, reduce air-conditioning energy consumption and provide and maintain a comfortable visual environment indoors [1]. Liquid crystals (LCs) offer an excellent material system for developing smart windows, and applications such as switchable privacy glass [2], solar heat rejection [3] and switchable color [4] have been demonstrated. Various LC technologies based on light scattering or reflection have been examined to create smart windows for regulating indoor solar irradiation, especially for visible light and infrared (IR) light. Visible light is used to maintain interior illumination levels, while IR light is responsible for indoor temperature control.

Various technologies are available for controlling visible light, including LC molecules mixed with ionic dopants or a guest LC material, which can be used to switch between the haze-free transparent and high-haze opaque states owing to the electrohydrodynamic effect [5,6,7]. Furthermore, technologies based on light scattering using LC/polymer composites, such as polymer stabilized LCs (PSLC) and polymer dispersed LCs (PDLC), have been developed to control visible light [8,9,10,11,12]. Smart windows have been developed using smectic/cholesteric liquid crystal (CLC) phases, where the window can be reversibly switched between transparent (planar and homeotropic) and scattering (focal conic) states [13,14]. For technologies controlling IR light, polymer stabilized CLC (PSCLC) has been widely studied [15,16,17]. CLC are nematic liquid crystals (NLC) that contain a chiral component that generates a helical twist between LC layers, thus allowing the selective reflection of circularly polarized incident light of the same handedness as its helix. The wavelength of light reflected by the CLC is determined by its helical pitch (p). The bandwidth of the reflected light is defined as $∆\lambda \approx ∆n\times p=(n_{e}-n_{o})\times p$, where ∆n is the birefringence of the host LC, and $n_{e}$ and $n_{o}$ are extraordinary and ordinary refractive indices, respectively [18,19]. For PSCLC-based IR reflectors, the IR reflection band can be electric-regulated dynamically, which is attributed to pitch distortion caused by the displacement of the polymer network under an external E-field [15,17,20]. We previously reported an IR reflector that can reflect a broad band of IR light from 725 to 1435 nm dynamically upon application of the E-field, while maintaining high transmittance in the visible region [17]. An IR reflector of this type could be used as a smart window, allowing solar IR energy to enter during the winter while reflecting it during the summer, reducing energy consumption for heating and cooling energy demands in the built environment.

Although much effort has been dedicated to developing LC-based smart windows for regulating visible and IR light, the ability to regulate both types of light is extremely desirable. Du et al. demonstrated an LC based window that can be switched between transparent and opaque for visible light through a voltage pulse while maintaining a 220 nm reflection bandwidth for near IR light [21]. However, the fabrication process is complicated, involving the construction of a hybrid structure using CLC and chiral polymer film. Moreover, the reflection bandwidth of the IR light is narrow and cannot be dynamically regulated, indicating that it will always reject solar heat, regardless of whether it is winter or summer.

In this work, we demonstrate an electrically switchable window that can dynamically regulate both visible and IR light. The window is based on PSCLC sandwiched between two transparent glass substrates. The PSCLC has three terminal electrodes—one plate electrode on the top substrate, and the other two interdigitated electrodes on the bottom substrate. When an alternating current (AC) E-field is applied between the top and bottom electrodes, the PSCLC switches from a transparent cholesteric state to scattering focal conic state, reducing transmittance from 90% to less than 15% in the whole visible region. When a direct current (DC) E-field is applied between the interdigitated electrodes, the non-uniform E-field distribution leads to helix pitch distortion of the CLC, thus resulting in a broadband reflection. The IR reflection bandwidth can be dynamically tuned from 158 to 478 nm. By adjusting the E-field intensity, the transmittance in the visible range and the IR reflection bandwidth can both be controlled, and after the E-field is removed, both can be returned to initial conditions. This attractive feature of the window makes it possible to perform indoor light and heat management on demand. Such windows would be an extension of the currently available smart windows, with considerable potential for practical applications in green buildings and automobiles.

## 2. Materials and Methods

### 2.1. Materials

The PSCLC mixture was prepared by mixing 81.4 wt% of a nematic LC E7 (The ∆n of E7 is 0.217 at λ = 589 nm and T = 25 °C, and the Δε of E7 is 11.4 at 1 kHz and T = 25 °C; Merck, Darmstadt, Germany), 13.8 wt% of a right-handed chiral dopant S811 (Merck), 3.0 wt% of a diacrylate monomer (RM257, Merck, Darmstadt, Germany) and 1.0 wt% of photo-initiator Irgacure-651 (Ciba Specialty Chemicals (China) Ltd., Shanghai, China). All ingredients were used without further purification. Figure 1A shows the chemical structures of S811, RM257 and Irgacure-651. To enable the reflection of IR light, the chiral dopant concentration was selected such that the reflection notch of the PSCLC cells was centered at 1050 nm.

### 2.2. Sample Preparation

The interdigitated electrodes (Wuhu Changxin Technology Co., Ltd., Anhui, China) and the plate electrode glass were used as the bottom and top substrates to make the cell. Figure 1B shows the two-dimensional top view of the interdigitated electrodes pattern. The width of the electrode is 95 µm, and the gap between the interdigitated electrodes is 25 µm. The electrode and gap are periodically distributed. Both the top and bottom substrates underwent a 10 min ultrasonic cleaning process in acetone, isopropanol, and deionized water, followed by a 20 min UV-ozone treatment (UV Products, BZS250GF-TC, Shenzhen, China) to make them more hydrophilic. The polyvinyl alcohol (PVA) alignment layer was spin-coated on both substrates. Then, the substrates with the PVA layer were placed parallel to each other in the rubbing direction and glued together with UV curable glue containing a 30 µm spacer (SiO_2_). After 1 min of UV irradiation, the LC cell with a cell thickness of 30 µm was obtained, as shown in Figure 1C. To examine the impact of interdigitated electrodes with different gaps on the performance of the sample, interdigitated electrodes with a 95 µm electrode width and various gap sizes (25, 45, 65, 85 and 105 µm) were used. Moreover, for investigating the impact of interdigitated electrodes with different electrodes on the performance of the sample, interdigitated electrodes with a gap width of 25 µm and different electrode widths (35, 55, 75, 95 and 115 µm) were used. The LC mixture was stirred and filled into the cell by capillary force. The cells were then exposed to UV light of 365 nm (28 mW/cm^2^) to polymerize. The whole preparation process was conducted in the yellow-light area.

### 2.3. Characterization

The optical characterization of the sample cells was conducted using unpolarized spectrophotometry (Lambda 950, PerkinElmer, Shanghai, China) in transmission mode at normal incidence to obtain the spectra of the sample cells [16]. A bare ITO glass substrate was used for the baseline correction of the instrument prior to the measurement of the photoelectric properties. Optical microscope photographs of PSCLC sample cells were observed at room temperature using a polarizing light microscope (POM, LEICA DM2700P, Leica, Solms, Germany) to analyze changes in the liquid crystal texture of the sample cells [16]. The device was driven by a single-phase AC power source with a frequency of 50 Hz, which generates voltages from 0 to 300 V with a power factor of 0.8, and a DC power source that generates voltages from 0 to 210 V.

## 3. Results and Discussion

Firstly, an AC E-field was applied between the top and bottom electrodes. Because of the small gap (25 µm) of the interdigitated electrodes on the bottom substrate, an approximated vertical E-field was created in the sample cell (Figure 2(Ai)). In the gap region, the E-field was slightly inclined. Under this E-field, the alignment of the LC molecules changed from a cholesteric texture (Figure 2(Aii)) to a focal conic texture (Figure 2(Aiii)), which is evidenced by the POM images shown in Figure 2(Bi,Bii). Accordingly, the sample switched from transparent to opaque because of the realignment of the LC molecules. As shown in Figure 2(Biii,Biv), under the vertical E-field, the background patterns under the sample could not be observed, thus confirming a scattering opaque state. Figure 2C shows the transmittance spectra of the sample under different AC voltages. At the off state (0 V/µm), the transmittance in the entire visible region was greater than 90%, and there was a reflection band centered at 1050 nm due to the Bragg reflection of the cholesteric phase. In the on state, the transmittance decreased as the applied voltage increased. For example, under the voltage of 3.34 V/µm, the transmittance dropped to less than 15% in the visible region (from 400 to 760 nm). Moreover, the reflection band disappeared under high voltage, which was due to the randomly distributed helical axis of the focal conic state [13]. It could be noticed that the operational voltage was high, which is attributed to two reasons. One is that the inclined E-field in the gap regions of the interdigitated electrodes cannot completely act on rotating the LC molecules with positive dielectric anisotropy (E7, ∆ε=11.4). The horizontal component of the inclined E-field actually prevents the LC molecules from rotating. The other reason is the anchoring force of the polymer networks among the LC molecules. When removing the AC voltage, the alignment of the LC molecules goes back to cholesteric texture because of the anchoring force of the polymer network. Therefore, the sample will revert to its initial transparent state.

When a DC voltage is applied between the interdigitated electrodes, the sample remains transparent, as shown in Figure 3(Ai,Aii). Regardless of whether the voltage is applied, the background patterns under the sample are clearly visible, indicating a good transparency state. The POM images (Figure 3(Aiii,Aiv)) indicate that the alignment of CLC molecules remains in the cholesteric texture under the DC voltage. Figure 3B shows the transmittance spectra of the sample under different DC voltages, thus exhibiting an electrically tunable bandwidth broadening. The reflection band is centered at 1050 nm with an initial bandwidth of 158 nm. When a DC voltage is applied, the bandwidth broadening becomes almost symmetric, extending simultaneously towards the red and blue sides compared to the original reflection notch wavelength. At the DC voltage of 2.76 V/µm, the sample shows a broad band of IR light from 800 to 1278 nm (with a reflection bandwidth of 478 nm), while predominantly remaining transparent in the visible region with high transmittance in the 400–760 nm range. When the DC interdigital E-field is removed, the alignment of the LC molecules goes back to cholesteric texture because of the anchoring force of the polymer network. Therefore, the reflection bandwidth will revert to its initial value of 158 nm.

The bandwidth broadening of the CLC can be attributed to the distortion of its helix pitch under an in-plane E-field between the interdigitated electrodes. Typically, applying a DC voltage between the interdigitated electrodes generates a non-uniformly distributed in-plane E-field in the sample cell [22]. Figure 3(Ci) shows the in-plane E-field configuration over a spatial period. In the region of the interdigitated electrodes, the E-field is perpendicular to the substrate (E_1_ in Figure 3(Ci)), while in the gap region between the neighboring electrodes, the E-field is parallel to the substrate (E_2_ in Figure 3(Ci)). The parabola-like distribution of the in-plane E-field induces a complicated orientation change of the CLC molecules. In particular, in the gap region, the E_2_ is perpendicular to the helical axis of the CLC, which results in the unwinding of the CLC helix. Therefore, the helical pitch p will be elongated [22]. Moreover, E_2_ is not homogeneous in the gap region, where the E-field close to the bottom substrate is larger than that close to the top substrate. As a result, the unwinding of the CLC helix is not uniform, and the pitch varies along the cell thickness in the samples. Accordingly, the reflection peak of the CLC will be shifted to the red side with band broadening. In the region of the electrodes, the DC E-field (E_1_) is parallel to the helix axis of the CLC. The polymer network traps cations by electrostatic force [23,24,25]. Upon applying the DC E-field (E_1_), an electromechanical force is exerted on the polymer network because of the trapped ions, leading to the deformation of the polymer network. Because of the aligning effect of the polymer network on the CLC molecules, the deformation results in a non-uniform pitch distribution by stretching the pitch near the anode side and compressing the pitch near the cathode side [26], as shown in Figure 3(Ciii). Consequently, the reflection band is broadened to include both the blue and red sides (Figure 3B). It is noted that, with increases in the applied DC voltage, the transmittance of the sample in the visible region slightly decreases. For instance, without applying the DC voltage, the transmittance in the visible region is above 90%, and it decreases to less than 80% at a high voltage of 2.76 V/µm. The decrease in the transmittance in the visible region is likely attributed to the scattering resulting from the focal conic state at such a high E-field.

Next, we examined the effect of the interdigitated electrodes with different gaps on the device’s performance. We created the cell using interdigitated electrodes with a fixed electrode width of 95 μm, but with various gaps ranging from 25 to 105 μm. 

After applying an AC voltage of 80 V between the top and bottom electrodes, we measured the transmittance spectra of the samples created using interdigitated electrodes with different gaps. Figure 4A displays the transmittance of the samples at 600 nm. The sample with a small interdigitated electrodes gap shows low transmittance. For example, when the gap was 25 μm, the transmittance was 40%. As the gap increased, the transmittance increased, rising to more than 60% when the gap was magnified to 105 μm. As previously discussed, when an AC E-field is applied between the top and bottom electrodes, the E-field is vertical to the substrate in the area of the interdigitated electrodes and inclined in the gap region. As the gap increases, the inclination angle of the E-field becomes larger, resulting in a smaller vertical component of the inclined E-field. Thus, a weaker vertical field acts on the CLC molecules in the gap region, making it more challenging to realign the CLC molecules from cholesteric texture to focal conic texture. Consequently, under the same applied AC voltage, the sample with the larger interdigitated electrodes gap results in a higher transmittance.

As mentioned earlier, a DC voltage applied between the interdigitated electrodes creates a non-uniformly distributed in-plane E-field that distorts the CLC helix pitch, thus resulting in broadband reflection. The different gaps of the interdigitated electrodes intrinsically affect the intensity of the in-plane E-field, which in turn influences the reflection bandwidth broadening of the sample. Hence, we measured the reflection band of the samples fabricated with various interdigitated electrodes gaps. Figure 4B shows the reflection bandwidth of samples under a DC interdigital voltage of 50 V. The reflection bandwidth decreases as the gap increases. For instance, at a gap of 25 μm, the reflection bandwidth of the sample is 387 nm, which gradually decreases to 313, 285, 252 and 233 nm when the gap increases from 25 to 45, 65, 85 and 105 μm. We attributed this decrease to the reduction in interdigital E-field intensity as the gap widens. In particular, in the sample with a small gap, the stronger in-plane E-field made it easier for the CLC helix in the gap region to be unwound. Furthermore, in the interdigitated electrodes region, the stronger E-field, which is parallel to the CLC helix, caused a larger displacement of the polymer network, creating a larger pitch gradient and resulting in a broader reflection bandwidth.

Furthermore, we investigated the effect of the interdigitated electrodes with different electrode widths on the device performance. We made the cell by using the interdigitated electrodes with the gap fixed at 25 μm, and various electrode widths ranging from 35 to 115 μm.

We measured the transmittance spectra of the samples under an AC voltage of 80 V applied between the top and bottom electrodes. The transmittance of the samples at a wavelength of 600 nm is shown in Figure 5A. The sample with the small electrode width showed low transmittance. For example, when the electrode width was 35 μm, the transmittance was 35%. As the electrode width increased, the transmittance increased and rose to 49% when the electrode width was increased to 115 μm. As we discussed above, when an AC E-field is applied between the top and bottom electrodes, an approximated vertical E-field is created in the sample cell. The electric field intensity between the capacitor’s two plates can be shown as $E=Q÷\left(\varepsilon_{0}\times S\right)$, where Q is the amount of charge in the capacitor’s plate, $\varepsilon_{0}$ is the dielectric constant of the material between the capacitor’s two plates (in this case, $\varepsilon_{0}$ is the dielectric constant of the PSCLC material), and S is the relative area of the two plates [27,28]. We can regard the cell as a capacitor, with the top substrate’s plate electrode and the bottom substrate’s interdigitated electrodes serving as its two plates. The top and bottom electrodes of the capacitor contain an equal quantity of charge when an AC field is applied between them. As the electrode width increases, the relative area of the capacitor’s two plates increases. Since the AC power supply and the PSCLC material are fixed, the samples created with interdigitated electrodes of various electrode widths have the same charge Q and dielectric constant $\varepsilon_{0}$. Thus, the E-field of the capacitor decreases as the relative area of the two plates increases. In other words, when the AC voltage is held constant, the E-field intensity in the cell decreases as the electrode width increases. In this sense, a weaker E-field acts on the CLC molecules in the cell, which makes it harder to realign the CLC molecules from cholesteric texture to focal conic texture. As a result, the sample with a larger electrode width produces a higher transmittance with the same applied AC voltage.

As we discussed above, a DC voltage between the interdigitated electrodes creates a non-uniformly distributed E-field, which causes a distortion of the CLC helix pitch, resulting in a broadband reflection. Different electrode widths of the interdigitated electrodes intrinsically influence the E-field intensity, which in turn affects the reflection bandwidth broadening of the sample. Therefore, we measured the reflection band of the samples fabricated with various electrodes. Figure 5B shows the reflection bandwidth of the samples under a DC interdigital voltage of 50 V. We can observe that the reflection bandwidth decreases as the electrode width increases. For example, at an electrode width of 35 μm, the reflection bandwidth of the sample was 451 nm, which gradually decreased to 447, 432, 386 and 371 nm when the electrode width enlarged from 35 to 55, 75, 95 and 115 μm. We attributed this to the reduction in interdigital E-field intensity when the electrode width increases. According to Gauss’s theorem, the electric field intensity generated by a charged plate can be calculated as $E=\sigma ÷{2\varepsilon }_{0}=\left(Q÷S\right)÷2\varepsilon_{0}$, where σ is the charge surface density, $\varepsilon_{0}$ is the dielectric constant, Q is the amount of charge and S is the area of the plate [29,30]. When a DC E-field is applied to interdigitated electrodes, we can regard the interdigitated electrodes as a charged plate. The equivalent charged plate area increases as the electrode width increases in the unit area. The DC power supply and the PSCLC material are fixed, so the amount of charge Q and dielectric constant $\varepsilon_{0}$ of the samples are the same, both of which were fabricated by using interdigitated electrodes with different electrode widths. Therefore, the charge surface density σ of the sample decreases as the electrode width increases. Furthermore, the electric field intensity decreases as the charge surface density decreases. Specifically, in the sample with small electrode width, the total electrode area per unit area was small and the E-field intensity was strong. The stronger E-field makes it easier for the CLC helix to have a non-uniform pitch distribution, resulting in a broader reflection bandwidth.

For the practical use of electrically responsive dual-function glass as a smart window, an important factor that needs to be considered is its energy consumption [17]. The electric power needed to operate the PSCLC was calculated to evaluate its power consumption. Figure 6A shows exponential decays in the current flow when operation AC voltages of 60, 70, 80, 90 and 100 V were applied. The current of the device reaches its steady state after nearly 200 s. The steady current increases as the operational voltage increases. Driven by an AC voltage of 100 V, the steady current is 9.076 μA. The power consumption can be calculated as P=V×I, where V is the applied voltage and I is the current at steady state [17]. The effective area of the PSCLC is 1.6 × 2.0 cm^2^. The power factor of the single-phase AC power source is 0.8. Accordingly, at 100 V AC voltage, the total power consumption to switch and maintain the PSCLC in an opaque state was around 2.3 × 10^3^ mW m^−2^. Figure 6B shows exponential decays in the current flow when DC voltages of 30, 40, 50, 60 and 69 V were applied. The current of the device reaches its steady state after nearly 240 s. Driven by a DC voltage of 69 V, the steady current is 0.227 μA. The corresponding power consumption to switch and maintain the device in a broad reflection state is calculated to be only 48.9 mW m^−2^.

## 4. Conclusions

After combining a top plate electrode substrate with a bottom interdigitated electrode substrate, we successfully fabricated a PSCLC smart window capable of regulating both visible light and infrared (IR) light by applying different electric fields. The interdigitated electrodes design enables the window to achieve dual functionality by creating different electric fields. In particular, a vertical alternating current (AC) electric field between the top and bottom substrate switches the CLC texture to a focal conic texture, thus resulting in a decrease in transmittance throughout the visible region from 90% to less than 15%. Furthermore, an in-plane interdigital direct current (DC) electric field leads to helix pitch distortion of the CLC, thus resulting in a broadband reflection. The reflection bandwidth in the IR region can be dynamically tuned from 158 to 478 nm. The electric field strength can be adjusted to regulate both the transmittance in the visible range and the IR reflection bandwidth. Once the electric field is removed, the window is restored to its initial state. This attractive feature of the window makes it possible to perform indoor light and heat management on demand. Such a window would be an extension of the currently available smart windows. However, the practical application of the LC smart window is limited by its stability under a strong electric field and under strong light flux. If this problem is solved or optimized, smart windows have great potential for practical applications in green buildings and automobiles.

## Figures and Tables

**Figure 1 polymers-15-01734-f001:**
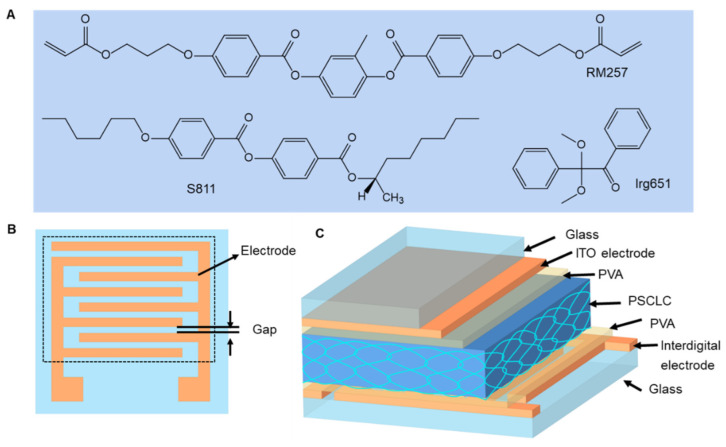
(**A**) The chemical structures of S811, RM257 and Irgacure-651. (**B**) Top view of the bottom glass substrate and the interdigitated electrodes pattern (the analyzed responsive region is indicated by the dashed square). (**C**) Three-dimensional structure side view of the device.

**Figure 2 polymers-15-01734-f002:**
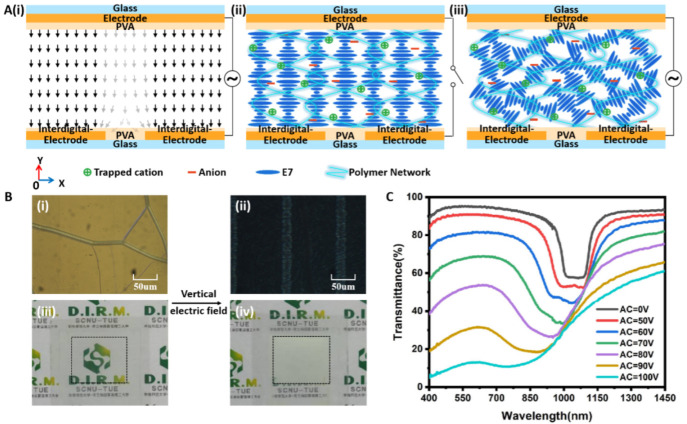
(**A**) Schematic diagram of (**i**) the vertical AC E-field between the top and bottom electrodes, (**ii**) the initial CLC cholesteric texture before applying the E-field, (**iii**) the focal conic texture after applying the E-field. (**B**) The POM images of the sample before (**i**) and after (**ii**) applying the E-field. The pictures of the sample before (**iii**) and after (**iv**) applying the E-field. (**C**) Transmittance spectra of the sample under different AC voltages.

**Figure 3 polymers-15-01734-f003:**
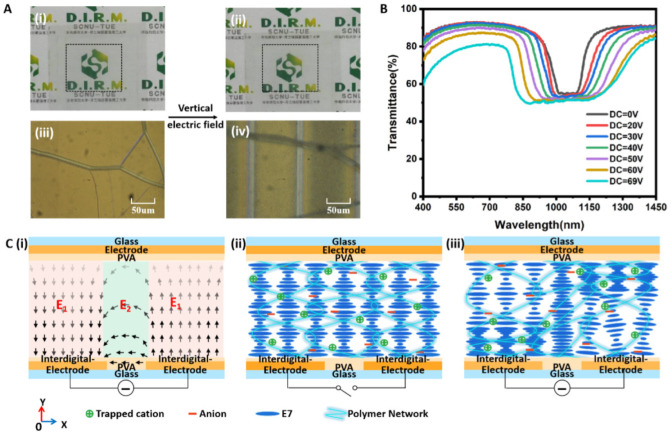
(**A**) The pictures of the sample before (**i**) and after (**ii**) applying the in-plane interdigital E-field. The POM images of the sample before (**iii**) and after (**iv**) applying the in-plane interdigital E-field. (**B**) Transmittance spectra of the sample under different in-plane interdigital voltages. (**C**) Schematic diagram of (**i**) the in-plane interdigital E-field between the neighboring electrodes, (**ii**) the initial CLC cholesteric texture before applying the E-field and (**iii**) the non-uniform pitch texture after applying the E-field.

**Figure 4 polymers-15-01734-f004:**
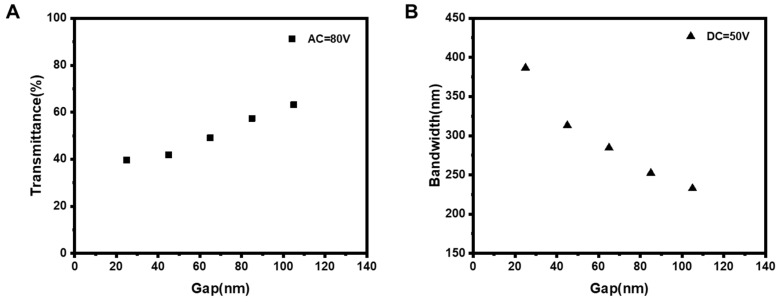
Samples are fabricated by using interdigitated electrodes with different gaps. (**A**) The transmittance (@600 nm) of the samples when applying an AC voltage of 80 V between the top and bottom electrodes. (**B**) The reflection bandwidth of the samples in the infrared region when applying a DC voltage of 50 V between the interdigitated electrodes.

**Figure 5 polymers-15-01734-f005:**
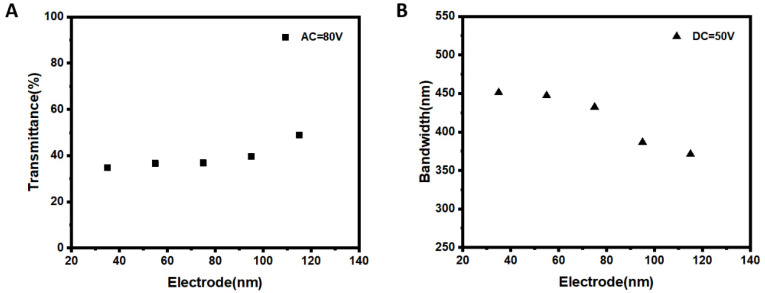
Samples are fabricated by using the interdigitated electrodes with different electrode widths. (**A**) The transmittance (@600 nm) of the samples when applying an AC voltage of 80V between the top and bottom electrodes. (**B**) The reflection bandwidth of the samples in the infrared region when applying a DC voltage of 50V between the interdigitated electrodes.

**Figure 6 polymers-15-01734-f006:**
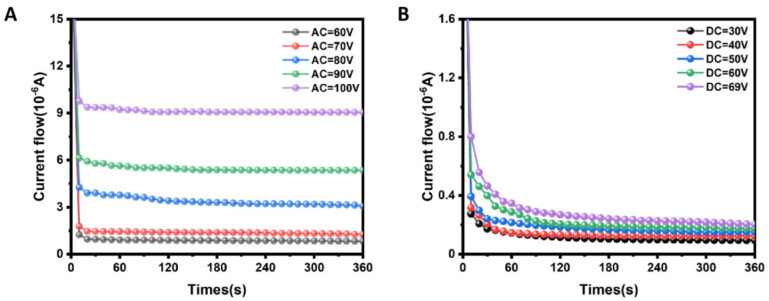
Samples fabricated by using the interdigitated electrodes with electrode width 95 µm and gap width 25 µm. (**A**) Current flow in the visible light scattering in the presence of different operation voltages. (**B**) Current flow in the IR broadband reflection in the presence of different operation voltages.

## Data Availability

Data presented in this study are available on request from the corresponding author.

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
