# Peer review of "Dual-Function Smart Windows Using Polymer Stabilized Cholesteric Liquid Crystal Driven with Interdigitated Electrodes"

_polymers, 2023, doi:10.3390/polym15071734_

Round 1

Reviewer 1 Report

This paper presents a new structure for electrically switchable windows capable of regulating both visible and infrared (IR) light by applying AC or DC voltage. By utilizing an electric field to switch the cholesteric liquid crystals (CLCs) between different states, the device's transmittance can be significantly tuned at wide wavelengths range. The authors also provide an explanation of the impact of AC/DC voltage on the CLC phase change. However, there are still a few aspects that need to be clarified before publication,

1.     The author did not provide information regarding the frequency of the AC source used in the experiment design. Please provide this information in the revised manuscript for the benefit of the readers.

2.     The authors presented the transmittance spectra of the device under AC voltage in visible light region in Figure 2C. However, it’s unclear how the AC voltage affects the transmittance of the device in the IR region. It is recommended that the authors include the transmittance spectra of the device under full wavelength like Figure 3B.

3.      The electric field values were presented in volts without normalization, which makes it difficult to compare the results with other works. It’s recommended that the authors normalize the E-filed values for better comparison.

4.     The authors provided an explanation for the mechanism of reflection band broadening in lines 162-204. It is suggested that the broadening results from two effects: 1) Non-uniform in-plane E-field between the interdigital electrode, causing the reflection band broadening to the red side; 2) The perpendicular DC electric field (E1) on top of the interdigital electrode deforms the polymer network, resulting in the reflection band broadening to the blue side. While the authors referenced Xianyu’s work[22] supporting the first explanation, a control experiment or reference to support the second explanation is missing.

To better support the hypothesis, we recommend that the authors provide a reference or conduct a control experiment to evaluate the reflection band broadening to blue side mechanism caused by the perpendicular DC electric field. This additional information will provide better evidence to support the authors' explanation of the broadening mechanism.

Reviewer 2 Report

Dear Authors,

The work is devoted to the development of a smart window concept built on a polymer-stabilized cholesteric LC cell. The novelty is connected with the use of interdigitated ITO electrodes to drive the LC cell. Also, the second electrode is solid ITO coating. The authors have observed the effect of CLC reflection bandwidth broadening after the application of strong offset voltage. Although the observed effect has some merits, the estimation of incident irradiation power in the scope of the smart window application was not performed. 

General comments:

1. As a matter of fact, ITO coating technology is widely used in windows for heat management. It is known as low-emission or Low-E windows. The ITO coating strongly absorbs the light above the wavelength of 2 um. Such windows are characterized by a set of performance coefficients, such as Heat loss, Visible Light Transmittance, Solar heat gain and Light-to-solar gain. You can check it in this brief note: https://glassed.vitroglazings.com/topics/how-low-e-glass-works 

The reflection bandwidth in the NIR region is important and should be estimated as a portion of the total energy incident from the sun (see the Sun spectrum). Said efficiency compared to bare ITO glass would be of interest to the reader.

2. The high-voltage appliances are connected with high power consumption. The power consumption of the smart window should be estimated.

3. The stability concerns of organic material under strong light flux as well as under strong electric field are known to be critical for LCs. That is why inorganic solutions are widely applied in construction glasses, such as Low-E glass. 

Specific comments: 

Please, check the material parameters of E7 material – dielectric anisotropy should be larger according to [J. Li, C. H. Wen, S. Gauza, R. Lu and S. T. Wu. Refractive indices of liquid crystals for display applications, J. Disp. Technol. 1, 51-61, 2005.].

Figure 3B, 4A – please, add a scale bar to photos.

‘Interdigitated electrodes’ is a common term, please, use it instead of ‘interdigital electrode’.

Lines 18, 21, 22, 25 and through the text – ‘electric field’ instead of ‘filed’,

Line 20 - decreases,

Line 34 – attention.

Round 2

Reviewer 2 Report

The corrected manuscript can be published after minor text editing.

Thank you!

Author Response

请参阅附件。
